# Buccal Swab Samples from Japanese Brown Cattle Fed with Limonite Reveal Altered Rumen Microbiome

**DOI:** 10.3390/ani14131968

**Published:** 2024-07-03

**Authors:** Kentaro Harakawa, Shinpei Kawarai, Kirill Kryukov, So Nakagawa, Shigeharu Moriya, Kazuhiko Imakawa

**Affiliations:** 1Research Institute of Agriculture, Tokai University, Kumamoto 862-8652, Kumamoto, Japan; 3mtld001@mail.u-tokai.ac.jp (K.H.);; 2Center for Genome Informatics, Joint Support-Center for Data Science Research, Research Organization of Information and Systems, Mishima 411-8540, Shizuoka, Japan; 3Bioinformation and DDBJ Center, National Institute of Genetics, Mishima 411-8540, Shizuoka, Japan; 4Department of Molecular Life Science, Tokai University School of Medicine, Isehara 259-1193, Kanagawa, Japan; 5Micro/Nano Technology Center, Tokai University, Hiratsuka 259-1292, Kanagawa, Japan; 6Institute of Medical Sciences, Tokai University, Isehara 259-1193, Kanagawa, Japan; 7Photonics Control Technology Team, Riken Center for Advanced Photonics, Numazu 410-8601, Shizuoka, Japan

**Keywords:** Kumamoto strain of Japanese brown cattle (JBRK), Aso limonite, rumen metagenome, microbiota, correlation analysis

## Abstract

**Simple Summary:**

Farmers in the Mount Aso area, a southern district of Japan, raise the Kumamoto strain of Japanese Brown (Japanese brown) cattle on its grassland and often supplement their feed with Aso limonite, a mineral derived from deposited volcanic ash and magma, particularly for pregnant animals. However, the mechanisms associated with limonite’s beneficial effects have not been characterized. In this analysis, groups of Japanese brown pregnant cattle were either fed with limonite or without (control), and buccal swab samples were collected every 30 days for 90 days. DNA extracted from buccal swab samples was then subjected to a 16S rRNA gene sequence analysis, identifying core-rumen, non-core-rumen, as well as oral, bacteria. Limonite feeding reduced the proportion of core-rumen bacteria focused primarily on roughage digestion, instead promoting the non-core-rumen microbiome.

**Abstract:**

The areas of the Mount Aso grasslands in Kumamoto, Japan, are the primary location for the breeding of the Kumamoto strain of Japanese Brown cattle (JBRK). Although Aso limonite, deposited by volcanic ash and magma, has been commonly fed to pregnant JBRK in this area, the mechanisms of its salutary effects on pregnant JBRK have not yet been elucidated. Approximately 100 days before the expected day of calf delivery, seven JBRK (four supplemented with limonite and three controls without limonite) were assigned to this study, from which a buccal swab was collected at the highest rumination every 30 days for 90 days. DNA extracted from these swabs was then analyzed using a 16S rRNA gene amplicon sequence analysis. Statistically significant differences between the two groups were discovered through beta-diversity analysis, though results from alpha-diversity analysis were inconclusive. The microbiota identified were classified into six clusters, and three of the main clusters were core-rumen bacteria, primarily cellulose digestion in cluster 1, oral bacteria in cluster 2, and non-core-rumen bacteria in cluster 3. In the limonite group, core-rumen bacteria decreased while non-core-rumen bacteria increased, suggesting that limonite feeding alters rumen microbiota, particularly activation of non-core-rumen microbiota.

## 1. Introduction

Sustainable farming is desirable all over the world; however, continued demand for meat and other animal products necessitates livestock farming, which depends heavily on resources such as land, water, and feed. Farmers in the areas of the Mount Aso grasslands in Kumamoto prefecture, a southern district of Japan, have been using a Korean native breed as draft cattle for the last 1000 years. These breeding cattle are traditionally raised in various areas of 33,000 ha of Aso grassland, located on the top of the Aso caldera, during Summer and Autumn while barns located at the bottom of the caldera are used for Winter and Spring, including calf delivery. Approximately 100 years ago, the cattle improvement program started whereby European breeds including Simmental were introduced to these local cattle populations. This program continued and a consistent brown coat color was achieved, and the Japanese brown Kumamoto strain (JBRK) cattle registry association was formed in 1946 [1]. During the 1970s, the number of JBRK cattle exceeded more than 88,000 head [2], but now as the number of successors is becoming smaller and many livestock farmers are aging, the number of JBRK cattle is decreasing and the use of the Aso grassland areas has dwindled by 30% during the last 30 years [3]. The current total of JBRK constitutes only 20,000 heads, most of which are maintained in Kumamoto. It is believed that compared to JB, JBRK is better suited for grazing on pastures where grasslands are rich, as in the Aso area of Kumamoto [4].

In the soil of the Aso volcanic caldera, there exist large concentrations of an iron ore known as limonite, which was deposited from the iron-rich water that accumulated over time in shallow marshes and lakes. Limonite is composed of inorganic materials such as iron, silica, aluminum, calcium, sulfur, magnesium, potassium, and other minerals, and organic compounds such as carbohydrate, protein, and fat, the most abundant one being iron (Fe_2_O_3_) [5]. It was found that limonite can remove hydrogen sulfide from waste sewage sludge [2], and that limonite reduces methane emissions during ruminant in vitro fermentation [5]. It was also found that limonite supplementation ameliorates glucose intolerance in diabetic and obese mice [6]. These authors demonstrated that the limonite’s beneficial activity could be explained by changes in the gut microbiome under obese conditions in vivo [7]. This compound, popular as a feed additive in cattle, either powdered or in blocks containing salt and limonite, is rather widely used in Japan, however; its effects have not been well characterized.

Through worldwide efforts, a core-rumen microbiome has been determined to persist across a wide geographical range [8]. Accumulated data indicate that rumen microbial composition correlates with host productivity such as feed efficiency [9], milk composition [10,11], and even methane production [12,13]. Although rumen microbe sampling has traditionally been performed by directly examining the rumen contents, recent studies have shown that buccal swab samples from ruminants can serve as a reliable substitute for assessing rumen microbial populations [14,15,16,17,18].

Although the effect of limonite addition to pregnant cattle feed is being realized by local cattle farmers, the in vivo effect of this compound has not been characterized. We hypothesized that the beneficial effect of limonite feeding is due to changes in rumen microbiota. In this report, therefore, changes in rumen microbiota upon limonite feeding were tested through the use of buccal swab samples in late-pregnant JBRK cattle.

## 2. Materials and Methods

### 2.1. Animals, the Sampling Schedule, and Animal Ethics

It is commonly believed by Aso cattle farmers that limonite supplementation facilitates improved maternal outcomes such as easier calving and faster recovery after delivery, although there are as yet no peer-reviewed studies to provide support for these beliefs. We thus used a typical breeding farm (Kumamoto Aso Kenminfarm Co., Ltd., Kumamoto, Japan), raising an average of 100 cows, which deliver 3–4 calves a month. Seven pregnant heifers or cows of JBRK cattle, expected to deliver calves within a month (10 December 2022–10 January 2023), were chosen and subjected to the study. The cattle (46.3 ± 32.1 months old) were randomly divided into limonite (*n* = 4; ages of 23, 34, 37, and 63 months) and control without limonite (*n =* 3; 24, 51, and 92 months) feeding groups, and the buccal swab samples were collected from each cattle once a month (days 30, 60, and 90) after the beginning of the limonite feeding. The experiment was started on 1 September 2022, and was scheduled to complete at the 90-day mark about 10 days before the expected date of calving. The cattle were each fed whole-crop silage (10–13 kg/day) made from rice grown in the Aso area of Kumamoto, as well as commercial concentrate feed made of corn and barley (2.5 kg/day), twice a day at 8:00 h and 17:00 h. In the limonite group, the cattle were each fed 50 g/day limonite (Japan Limonite Co., Ltd., Kumamoto, Japan) mixed with the concentrate feed. The animals had free access to water. Thus, the rearing conditions and environment were the same except for the limonite feeding.

This study was performed in accordance with the guidelines for the care and use of experimental animals at Tokai University and the Act on Welfare and Management of Animals issued by the Ministry of the Environment, Government of Japan. The collection of buccal swabs and management of JBRK cattle were approved by the ethics committee of animal experiments at Tokai University (approval number 221075).

### 2.2. Collection of Buccal Swabs and DNA Preparation

The procedures for buccal swab collection and DNA preparation were performed according to the methods previously described [18]. Briefly, the buccal swabs were collected at peak rumination, four–five hours after the morning feeding. Two sterile cotton swabs (4.7 mm diameter), placed at both edges of a 30 cm stick, were inserted into the oral cavity between the back teeth and the inner cheek and were gently swabbed along the back teeth for approximately 5 s. The buccal swabs were then placed in a sterile 15 mL polypropylene tube containing nucleic acid preservation buffer (2 mL) [19] and the sampling was repeated with the other side of two cotton swabs, which were placed into the second tube in the same way. Four cotton swabs per animal were stored on ice during the 2 h transfer to our laboratory, and the samples were then stored at −80 °C until use.

Microbial DNA was extracted from four cotton swabs per cattle using the repeated bead beating plus column method [20]. In brief, 15 mL tubes were centrifuged, from which the preservation buffer was discarded. The cotton swabs and the precipitates were then vortexed and incubated at 70 °C for 3 min with total 0.4 g of the sterile glass beads, diameters 0.1 mm (0.3 g) and 0.5 mm (0.1 g), in a 0.8 mL cell lysis buffer (500 mM NaCl, 50 mM Tris-HCl, pH 8.0, 50 mM EDTA, and 4% sodium dodecyl sulfate). After centrifugation, the lysates were then homogenized at 550 rpm for 3 min with bead-beating equipment (ShakeMaster Auto, Biomedical Science, Tokyo, Japan). The bead beating was repeated with 0.2 mL cell lysis buffer, and all supernatants (total 1.0 mL lysate) were collected. The total DNA was precipitated from the supernatants with 10 M ammonium acetate and isopropanol. DNA was extracted by the use of a QIAamp Fast DNA Stool Mini Kit (Qiagen, Hilden, Germany) according to the instructions provided.

### 2.3. 16S rRNA Gene Amplicon Sequence

The V3-V4 regions of bacterial and archaeal 16S rRNA were amplified using the 341F (5′-AATGATACGGCGACCACCGAGATCTACACTCTTTCCCTACACGACGCTCTTCCGATCTCCTACGGGAGGCAGCAG-3′) and 805R (5’-CAAGCAGAAGACGGCATACGAGATNN-NNNNGTGACTGGAGTTCAGACGTGTGCTCTTCCGATCT-3’) primers [21], from which barcoded amplicons were paired-end sequenced on 2 × 301 bp cycle using the MiSeq platform (Illumina, SanDiego, CA, USA) and MiSeq Reagent Kit version 3 (600 cycles). These procedures were executed at a commercial laboratory (TechnoSuruga Laboratory Co., Ltd., Shizuoka, Japan). Paired-end sequencing reads were merged and processed through QIIME2 (version 2020.6) [22], and then representative sequences were generated using Divisive Amplicon Denoising Algorithm 2 (DADA2) [23]. The generated amplicon sequence variants (ASV) were assigned and annotated by using Greengenes Database version 13.8 by training a Naive Bayes classifier [24]. Annotation to the operational taxonomic units (OTUs) was performed on taxonomy 7 (species) levels.

### 2.4. Identification of Rumen-Specific Species

These QIIME2 merged contigs were compared to the previously established rumen bacteria [25], and the current bacterial taxonomy, from which only taxa classified at the species or genus levels were used for further analysis. The contig sequences were processed in the following ways: (a) Using BLASTN version 2.15.0+ with default parameters [26], all contigs were compared with the reference *Bos taurus* genome, and those that had hits with bit score of 100 or more were removed from the analysis. (b) The remaining contigs were compared using GSTK [27] and BLASTN and 18,941 representative prokaryote genomes were stored in the GenomeSync database [28] as of 20 February 2024. These representative genomes included all of the cattle rumen species and genera from Seshadri et al. (2018) [25]. According to the best BLASTN hit, taxonomic labels were then assigned to contigs, and the number of reads from each sample contributing to the known rumen bacterial taxa in cattle was counted.

The raw sequence data analyzed during the current study are available in the DDBJ Sequence Read Archive repository, accession number DRA018697 for the 16S rRNA gene sequence library.

### 2.5. Statistical Analysis

To justify the use of various months/ages of JBRK cattle, regression analysis was carried out to find the relation between ages and core-rumen bacteria, non-core-rumen bacteria, or oral bacteria, followed by a nonparametric Wilcoxon analysis to find difference in the relative abundance of core-rumen bacteria, non-core-rumen bacteria, or oral bacteria between limonite and control groups at each sampling day.

All samples from the control and limonite groups were compared by diversity and differential abundance analyses. The statistical significance between groups in the alpha diversity (Shannon’s diversity index and observed OTUs) was computed using the Kruskal–Wallis (KW) test and permutational multivariate analysis of variance. Beta diversity (Bray–Curtis, unweighted UniFrac, and weighted UniFrac distances) was visualized using Emperor tool. The Emperor tool was used to visualize principal coordinates analysis (3D-PCoA) plots. The 2D-PCoA was also performed using the qiime2R ver 0.99.13 [29] and tidyverse ver 1.2.1 [30] libraries in R [31]. The group significance was analyzed using ANOSIM. *p*- and q-values < 0.05 were considered statistically significant.

Read counts were filtered out if count was less than 10 and standard deviation was 0. Tag Count Comparison (TCC) baySeq [32] was employed for normalization and differential abundance analysis of the 16S rRNA sequencing and PICRUSt2 data. The TCC package was generated from original TbT methods (TMM-baySeq-TMM pipeline), consisting of a combination of the trimmed mean of M values (TMM) normalization [33] in edgeR [34] and annotated taxa detection in baySeq [35]. In this strategy, normalization of count data is iterated to avoid false positives; the method repeats this cycle three times [36].

Using the Tag Count Comparison (TCC) baySeq [26], bacterial data were further processed to obtain differential abundance, heatmap chart, and correlation network analyses. Community detection was performed with fastgreedy.community function of igraph. The generated community members and the quantified information were visualized as a heatmap chart using R package “gplots” with correlation coefficients between −2.0 and 2.0.

Correlation network analysis was performed [37] as follows: normalized data matrix was used for self-correlation analysis with R package “psych”. We defined the condition for significant correlation between all ASV pairs as R ≥ 0.3 and *p*-value < 0.05, followed by constructing an undirected network by using the R package “igraph”. Correlation network analysis was again performed using the microbiome data of control or limonite-fed cattle separately.

## 3. Results

### 3.1. Experimental Animals and 16S rRNA Gene Amplicon Sequence

A total number of 11 and 8 buccal swab samples from limonite and control cattle, respectively, were used. Two samples on day 90 were not collected because the heifer in each group had delivered the calf more than two weeks prior to the expected day of calving.

The total read counts (mean ± SD read count per sample) of control and limonite groups were 201,427 (25,178  ±  3177/head/day) and 319,093 (29,008  ±  6587/head/day) for 16S rRNA gene sequences, respectively. The total OTU numbers (mean ± SD) identified in the control and limonite groups were 1194 (149.3 ± 30.0/head/day) and 2239 (203.5 ± 30.3/head/day), respectively. The rarefaction curves were constructed to ensure a sufficient sequencing depth (22,000) for evaluating the dominant microbiome.

The relative abundance of oral, core-rumen, as well as non-core rumen microbiota, was calculated (Appendix A). Bacterial taxa were divided into rumen bacteria (core-rumen bacteria and non-core-rumen bacteria) and oral bacteria, which were 21.6–63.9% (core; 8.3–49.9% and non-core; 11.1–19.3%) and 35.5–78.3%, respectively. Direct examination of rumen or oral bacteria was not made; thus, all bacteria detected were processed together.

In the regression analysis, the relative abundance of core-rumen bacteria, non-core-rumen bacteria, or oral bacteria was not affected by cattle ages. Similarly, a nonparametric Wilcoxon analysis demonstrated that the relative abundance of core-rumen bacteria, non-core-rumen bacteria, or oral bacteria did not differ significantly between limonite and control groups (Appendix A).

### 3.2. Diversity Analysis

Because the regression and Wilcoxon tests demonstrated that cattle ages or relative abundance of rumen bacteria did not differ between groups, these data were subjected to diversity analyses. In the alpha diversity of the Shannon index and observed OTUs, the microbiome diversity of the limonite group appeared less than that of the control, though no statistical significance was observed (Appendix A). In the beta-diversity analysis, however, significant differences were observed between the groups in Bray–Curtis (pairwise ANOSIM, R = 0.60, q < 0.001) distances (Figure 1), weighted UniFrac (pairwise ANOSIM, R = 0.44, q < 0.006), and unweighted UniFrac (pairwise ANOSIM, R = 0.38, q < 0.005) distances (Figure 2).

### 3.3. Taxonomic Analysis

The top three phyla (mean % of each sampling day in the control and limonite groups) were Firmicutes (45.4 and 30.0% on day 30, 36.9 and 30.5% on day 60, 32.5 and 25.2% on day 90), Proteobacteria (27.7 and 52.3% on day 30, 19.7 and 50.2% on day 60, 39.2 and 50.5% on day 90), and Bacteroidetes (20.1 and 12.4% on day 30, 25.7 and 12.8% on day 60, 23.3 and 15.6% on day 90) (Figure 3 and Appendix A). In the control group, Firmicutes and Bacteroidetes, both belonging to the core-rumen bacteria [8], constituted an average of 55% or more, whereas those in the limonite group totaled 40%. It should be noted that Proteobacteria existed at the 50% level in the limonite group, and that the ratios of all three bacteria were fairly consistent during the experimental period. Moreover, Archaea (mean ± SD) existed in the control group (0.34 ± 0.24%) and limonite group (0.24 ± 0.17%) throughout the experimental period (Appendix A).

At the genus level, *Bibersteinia* (9.69% on day 30, 14.9% on day 60, 18.70% on day 90), *Prevotella* (7.42% on day 30, 12.0% on day 60, 10.1% on day 90), and *Streptococcus* (6.59% on day 30, 4.43% on day 60, 7.87% on day 90) existed consistently in the control group throughout the experimental period. In the limonite group, *Bibersteinia* (15.10% on day 30, 20.07% on day 60, 8.21% on day 90), *Streptococcus* (11.48% on day 30, 7.54% on day 60, 6.13% on day 90), and *Moraxella* (9.77% on day 30, 8.53% on day 60, 12.74% on day 90) existed (Figure 4, Appendix A).

Among Archaea, five and eight taxa were found in the control and limonite groups, respectively, of which *Methanobrevibacter* existed throughout the experimentation and Methanomassiliicoccaceae vadinCA11 was found only in the control group (Appendix A).

### 3.4. Rumen Microbe Analysis: Heatmap Visualization

Rumen microbiome was classified into six main clusters; cluster 1 consisted of core-rumen bacteria [8] characteristic of cellulose digestion (Appendix A). In cluster 2, oral bacteria were found, while non-core-rumen bacteria were found in cluster 3. Non-core-rumen bacteria were reduced in the limonite group, and rumen bacteria in the limonite group increased, but oral bacteria were reduced in cluster 6 (Figure 5).

### 3.5. Rumen Microbe Analysis: Correlation Network Analysis

In the taxonomic analysis, differences in the rumen microbiota were found between control and limonite groups, from which each of the microbiome networks was constructed. Using hub microbe detection methods, rumen microbes that serve core and significant interactions were visualized through correlation network analysis (Figure 6). The microbiome community was divided into six clusters, of which clusters 1 and 2 corresponded to the abundance of taxa in the control and limonite groups, respectively, as indicated by heatmap visualization. In cluster 1, the analysis revealed three “hub nodes” (*Pyramidobacter* species, unidentified species of order Bacteroidales, and Desulfuromonadales), which were significantly more connected within the network than other nodes according to all the node parameters. Among the three “hub” nodes, Dethiosulfovibrionaceae *Pyramidobacter* species existed in the control group throughout the course of the experimentation, but these species were not present after 60 days in the limonite group (Table 1).

The results of the correlation networks were further processed to show those in the control or limonite group separately (Figure 7). Among 261 taxa in the control group and 405 taxa in the limonite group, the “hub” node was not found. In the control group, two major and eight minor clusters were found: 127 taxa in cluster 1, most of which belong to core-rumen bacteria; 129 taxa, consisting of those in clusters 2–10; and 5 taxa, consisting of clusters 1, 2, and 6 (Figure 7A). In the limonite group, there are five major and five minor clusters: 94 taxa in cluster 1, related to the core-rumen bacteria; 150 taxa in cluster 2 with oral bacteria; 118 taxa in cluster 3 with non-core-rumen bacteria; 28 taxa in cluster 2 and 6, having oral bacteria; 150 taxa, consisting of oral bacteria; 9 taxa in cluster 4; and 6 taxa in clusters 1, 2, and 5 (Figure 7B, Appendix A).

From the results of differential abundance analysis, 246 taxa (*p* < 0.05) were found, from which 212 taxa were found in the limonite and 41 taxa in the control group. However, no taxon was found in the limonite group, which belonged to cluster 1, while 14 taxa belonged to cluster 1 in the control group. The remaining 212 taxa in the limonite and 26 taxa in the control group belonged to clusters 2–6 (Appendix A).

In the order of Actinomycetales, 39 taxa existed from which *Dietzia* and *Arthrobacter* existed at the genus level (*p* < 0.013) in the limonite group (Appendix A). *Dietzia* utilizes iron (III) oxide [38], which activates *Arthrobacter*, degrading lignin [39].

## 4. Discussion

In this study, rumen bacteria were studied through the DNA extracted from buccal swab samples, a proxy for rumen contents. It has been debated whether direct sampling is still required to study the rumen microbiome; however, accumulated evidence suggests that buccal swab samples are sufficiently similar as to act as a less-invasive surrogate [16,18]. Miura et al. (2022) also reported that buccal swab samples excluding putative oral bacteria exhibit a nearly identical microbiome to cannulated rumen contents [18]. Our staff were trained at the place where Miura et al. obtained direct rumen as well as buccal swab samples, enabling accurate reenactment of the same procedures for sample collection. Because microbiota from the control JBRK cattle without limonite feeding are similar to those of Miura et al. (2022), and more importantly those of the core-rumen microbiota found over the world [8], we therefore submit that the buccal swab sample is sufficient for rumen microbiome examination, if the timing of sampling is consistent. We thus opted to take buccal swab samples at the active phase of regurgitation [17], four to five hours after morning feeding [16] in this barn environment, and identified rumen microbiome differences subsequent to the limonite feeding.

In addition to the small number of samples, the ages of cattle in the control and limonite groups differ. It is therefore crucial to show that data within a group are fairly homogenous if each sample data can be used as a replicate. To study such variation, regression and a nonparametric Wilcoxon test were conducted, showing data homogeneity within a group. Data were then subjected to diversity studies.

In this study, oral and rumen bacteria were not separately evaluated. Because rumination/regurgitation is an integral part of roughage digestion in ruminants, grasses/straw can be digested from both anaerobic fermentation and aerobic regurgitation. Within the rumen bacteria (Appendix A), the core-rumen bacteria, the primary rumen bacteria for VFA production, made up averages of 25.7 ± 21.8% and 11.6 ± 7.74% in the control and limonite groups, respectively. The remaining microbiota were classified mostly as oral bacteria, and archaea existed in minuscule numbers. The ratios of oral bacteria in our buccal samples were similar to those published previously [14,17]. *Prevotella*, *Fibrobacter*, and *Ruminococcus* appeared less in the limonite group than in the control group (Appendix A). In the limonite group, while the ratio of core-rumen bacteria reduced in favor of the non-core-rumen bacteria. In the order of Actinomycetales, 39 taxa existed from which *Dietzia* and *Arthrobacter* existed at the genus level (Appendix A). *Dietzia* is known to utilize iron (III) oxide [38], possibly from limonite. This event activates *Arthrobacter* (*p* < 0.013), which degrades lignin, a heterogenous, recalcitrant, aromatic polymer, cross-linking with cellulose and hemicellulose via ester linkage [39]. This lignocellulose degrading system may fill in those with reduced core-rumen bacteria in the limonite group.

Among the three “hub nodes” assessed by correlation network analysis (Figure 6), *Pyramidobacter* sulfur-reducing bacteria of the Dethiosulfovibrionacea family were discovered, which promote growth through hydrogen sulfide production [40]. In addition, *Pyramidobacter* is also known to affect VFA production as well as the metabolic efficiency of rumen bacteria [41]. Another bacterial order of note is Desulfuromonadales, which reduce sulfate using fatty acids as substrates. Desulfuromonadales are also known as iron-reducing bacteria [42]. In the limonite group, however, these bacteria in cluster 1 (core-rumen microbiome) reduced to half their usual levels while those in cluster 2 (oral) and cluster 3 (non-core-rumen) increased, the latter containing *Dietzia* and *Arthrobacter*, suggesting that limonite-fed JBRK had adjusted to or reestablished a new rumen microbiome, which facilitated sufficient lignin digestion and/or VFA production. Furthermore, since limonite is very rich in the Aso caldera, this compound could be widely used once the mechanisms associated with its effect are elucidated.

## 5. Conclusions

While rumen microbiota in the control group were similar to the core-rumen microbiome, changes in those with limonite feeding were detected. Based on differential abundance analysis, regardless of JBRK’s ages, the relative abundance of the core-rumen microbiota and non-core-rumen microbiota is altered by limonite feeding. Although elucidation of the mechanisms associated with limonite feeding on changes in rumen microbiota requires further examination, the commonly held belief of Aso farmers that limonite feeding is effective for pregnant animals is in part supported by the results of this study.

## Figures and Tables

**Figure 1 animals-14-01968-f001:**
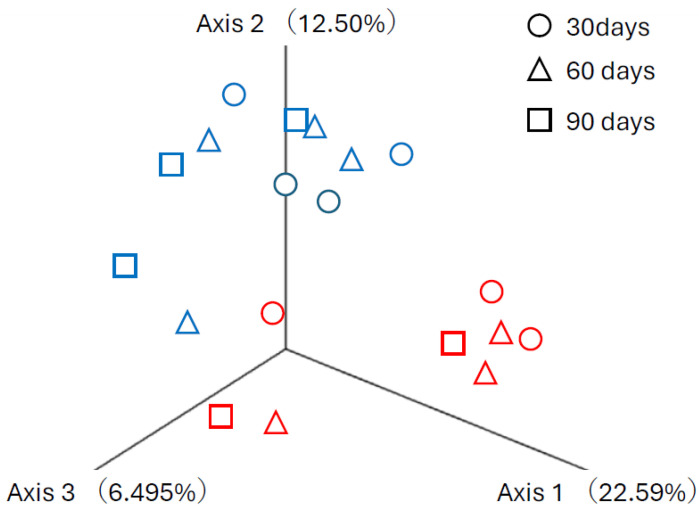
Beta diversity of buccal swab microbiome. Beta diversity is shown using 3D plots of Bray–Curtis distances. ○ 30 days, △ 60 days, □ 90 days following the initiation of limonite (blue) or not limonite (control, red) feeding.

**Figure 2 animals-14-01968-f002:**
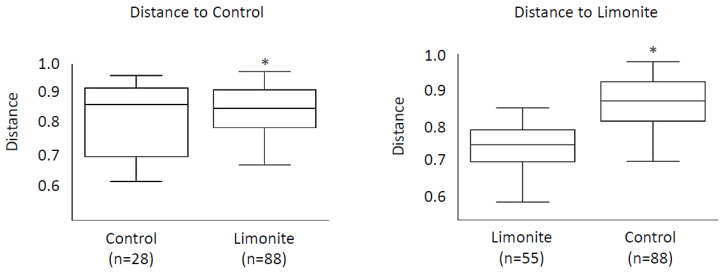
Beta diversity of buccal swab microbiome. Statistical differences in the structure between groups were detected by ANOSIM. Differences in groups of samples from one another were calculated using a permutation-based statical test. The ‘*n*’ corresponds to the calculation times that were performed at each group in the statistical analysis. Distance to Control is shown on the left and distance to limonite is presented on the right. * denotes statistical significance at *p* < 0.001.

**Figure 3 animals-14-01968-f003:**
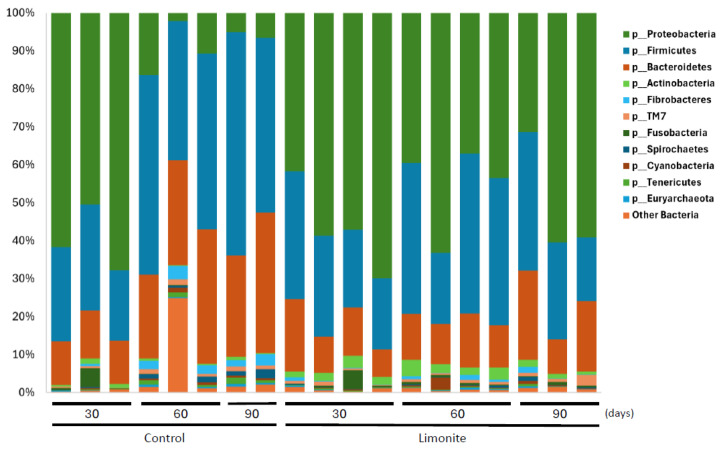
Relative abundance of bacterial taxa at the phylum levels. Bacterial taxa in the rumen microbe composition of each JBRK without (control) or with limonite feeding are shown.

**Figure 4 animals-14-01968-f004:**
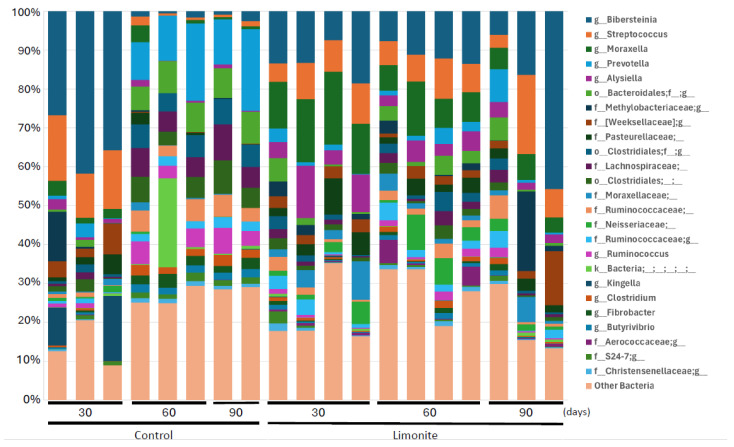
Relative abundance of bacterial taxa at the genus levels. Bacterial taxa in the rumen microbe composition of each JBRK without (control) or with limonite feeding are shown.

**Figure 5 animals-14-01968-f005:**
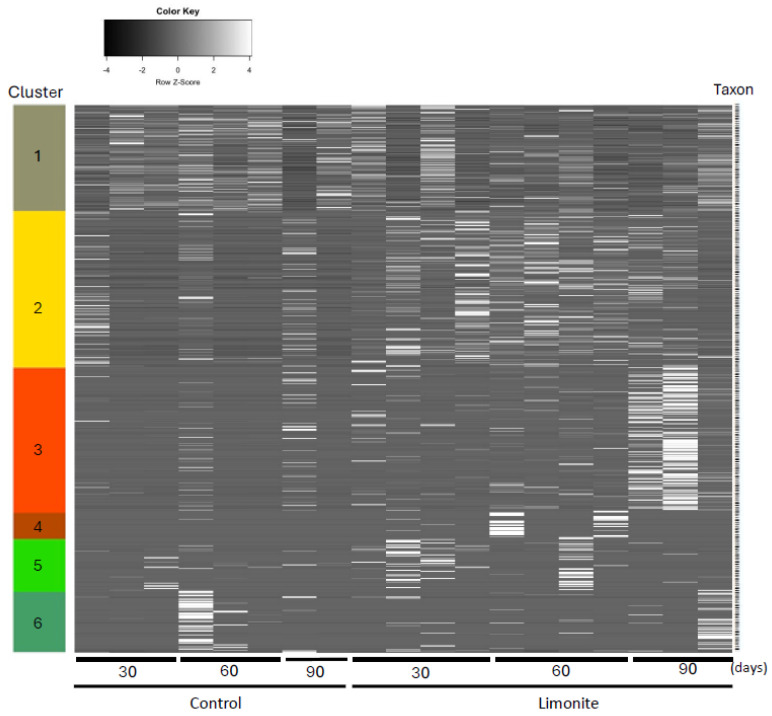
A heatmap chart of taxa abundance. Clusters 1 and 2 correspond to the abundance of taxa in the control and limonite groups, respectively.

**Figure 6 animals-14-01968-f006:**
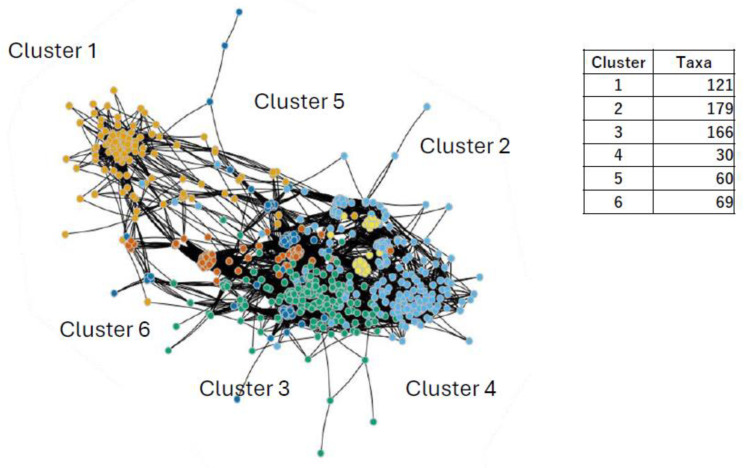
Correlation network analysis of rumen microbiota. A structural representation of the entire microbial community in the rumen. The structure is divided into 6 main clusters denoted by color (cluster 1, pale orange; 2, pale blue; 3, green; 4, yellow; 5, deep blue; 6, orange). In cluster 1, the three “Hub nodes” (denoted as squares) (*Pyramidobacter* species, and unidentified species of order Bacteroidales, and Desulfuromonadales) are significantly more connected within the network than other nodes.

**Figure 7 animals-14-01968-f007:**
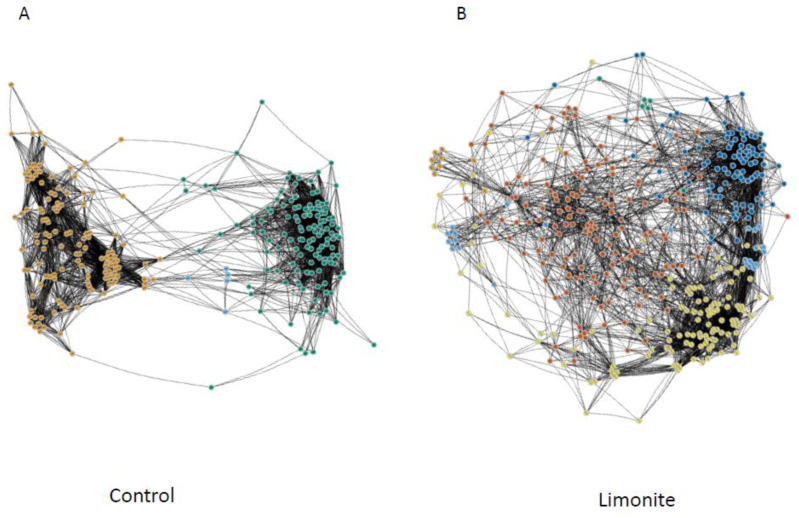
Correlation network analysis of rumen microbiota in the control or limonite group. A structure of the whole microbial community in the rumen of JBRK cattle fed (**B**) limonite or (**A**) not (control): (**A**) Among 261 taxa in the control group and 405 taxa in the limonite group, “hub” node was not found in this separate analysis. In the control group, two major and eight minor clusters were found: 127 taxa in cluster 1 (green), most of which belong to core-rumen bacteria; 129 taxa in pale orange, consisting of those in clusters 2–10; and 5 taxa in pale blue, consisting of clusters 1, 2, and 6. (**B**) In the limonite group, there are four major and six minor clusters: 94 taxa in cluster 1, related to the core-rumen bacteria (deep blue); 150 taxa in cluster 2 with oral bacteria (orange); 118 taxa in cluster 3 with non-core-rumen bacteria (yellow); 28 taxa in cluster 2 and 6, having oral bacteria (pale blue); 150 taxa, consisting of oral bacteria (orange); 9 taxa in cluster 4 (pale orange); 6 taxa in clusters 1, 2, and 5 (green).

**Table 1 animals-14-01968-t001:** Differential abundance analysis detected three “Hub node” in rumen microbiota of control and limonite-fed JBRK.

Taxa	Control			Limonite			
	30 (*n* = 3)	60 (*n* = 3)	90 (*n* = 2)	30 (*n* = 4)	60 (*n* = 4)	90 *(n* = 3)	*p*-Value
Desulfuromonadales	0.03 ± 0.02%	0.05 ± 0.01%	0.02 ± 0.02%	0.00 ± 0.00%	0.01 ± 0.01%	0.33 ± 0.29%	0.81
Dethiosulfovibrionaceae *Pyramidobacter*	0.03 ± 0.03%	0.03 ± 0.08%	0.02 ± 0.02%	0.00 ± 0.00%	0.02 ± 0.02%	0.33 ± 0.30%	0.51
Bacteroidales	0.50 ± 0.36%	0.93 ± 0.43%	0.52 ± 0.39%	0.29 ± 0.36%	0.34 ± 0.25%	0.34 ± 0.45%	0.51

Mean ± SD (%).

## Data Availability

The data present in this study are deposited to DDBJ accession number DRA018697.

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
