# Peer review of "Buccal Swab Samples from Japanese Brown Cattle Fed with Limonite Reveal Altered Rumen Microbiome"

_animals, 2024, doi:10.3390/ani14131968_

Round 1

Reviewer 1 Report (Previous Reviewer 2)

Comments and Suggestions for Authors

I believe the authors have appropriately revised their previous manuscript (animals-3005622). The current manuscript (animals-3079201) is suitable for publication in animals.

Author Response

Thank you very much for taking the time to review this manuscript.

Reviewer 2 Report (Previous Reviewer 3)

Comments and Suggestions for Authors

This study investigated the effects on the rumen microbiome of Japanese brown cattle fed with limonite. I see that the author has revised the content of the article. But there are still some things that need attention.

1. Line 32-33: The design of the experimental group and the selection of the sample were not rigorous.

2. Line 86: Did this experiment demonstrate the beneficial effects of limonite? Or are there other experiments where the addition of limonite has benefited Japanese Brown cattle?

3. Line 366-367: This is an unsupported inference because the article did not assess the physical condition of either group of animals.

4. Line 377: What specific bacteria have increased in abundance? What are their roles, and are they beneficial?

5. Line 408: Please check your references carefully. Page ranges for references are missing.

Author Response

Comments 1: Line 32-33: The design of the experimental group and the selection of the sample were not rigorous.

Response 1: We appreciate your comment. While we do understand your concern, the design and sample selection were based on the best practice available for this study conducted at a typical Japanese brown of Kumamoto (JBRK) strain farm with about 100 cattle, delivering calves 3-4 heads per month. It should be noted that we ensured our experimental design included controls, which accounted for variables such as diet except Limonite feeding, rearing condition, the months of pregnancy and health status of the animals to minimize bias. Even though the ages differed within limonite (n=4; ages of 23, 34, 37, and 63 months) and control (n=3; 24, 51, and 92 months) groups, the regression analysis demonstrated that relative bacteria abundances (oral, core rumen, and non-core rumen) were not affected by age. In addition, a nonparametric Wilcoxon analysis have revealed that relative bacteria abundances were not different between limonite and control groups at each sampling days. Based on these results, we consider each cattle as a replicate in this study and thus our findings statistically satisfactory, effectively elucidating the impact of Limonite feeding on the rumen microbiota during the late pregnancy of JBRK.

Currently, we are in the process of executing follow-up studies that will provide deeper insights into our results. These upcoming studies will incorporate more comprehensive biological and molecular analyses of blood characteristics, day 0 sampling and ruminal and intestinal flora, allowing us to validate our findings with enhanced rigor and a broader scope.

Comments 2: Line 86: Did this experiment demonstrate the beneficial effects of limonite? Or are there other experiments where the addition of limonite has benefited Japanese Brown cattle?

Response 2: Although the core-rumen microbiota reduced in favor of the non-core rumen bacteria, increase in Dietzia and Arthrobacter at the genus level were found in the limonite group (Supplement Table 5), suggesting that limonite may have enhanced lignin digestion. Please note that this is the first report on the effects of limonite on JBRK cattle.

Comments 3: Line 366-367: This is an unsupported inference because the article did not assess the physical condition of either group of animals.

Response 3: This was our mistake in that physical condition was not assessed in this experiment and thus such description was deleted from the text.

Comments 4: Line 377: What specific bacteria have increased in abundance? What are their roles, and are they beneficial?

Response 4: During the revision process, we found the following:  In the order of Actinomycetales, 39 taxa existed from which Dietzia and Arthrobacter existed at the genus level (Table S5). Dietzia is known to utilize iron (III) oxide (Wang et al. 2021), possibly from limonite. This event activates Arthrobacter (p<0.013) which degrades lignin, a heterogenous, recalcitrant, aromatic polymer, cross-linking with cellulose and hemicellulose via ester linkage (Couger et al. 2020). This lignocellulose degrading system may fill-in those with reduced the core-rumen bacteria in the limonite group.  We should have identified this before the original submission.

Comments 5: Line 408: Please check your references carefully. Page ranges for references are missing.

Response 5: Thank you very much for your comments. References were carefully evaluated and missing pieces are included in the revised manuscript.

Reviewer 3 Report (Previous Reviewer 4)

Comments and Suggestions for Authors

After reviewing the manuscript entitled “Buccal Swab Samples from Japanese Brown Cattle Fed with Limonite Reveal Altered Rumen Microbiome”, the following suggestions were made it. The manuscript is interesting and provides novel information on the use of Limonite in Japanese Brown Cattle. The authors have made an effort to accommodate each of my corrections in the initial version of the manuscript.

In its current form, the manuscript is suitable for publication.

Comments on the Quality of English Language

English only requires moderate changes

Author Response

Thank you very much for taking the time to review this manuscript.

This manuscript is a resubmission of an earlier submission. The following is a list of the peer review reports and author responses from that submission.

Round 1

Reviewer 1 Report

Comments and Suggestions for Authors

Please find comments in the file.

Reviewer 2 Report

Comments and Suggestions for Authors

In this paper (animals-3005622), the authors investigated the effects of limonite on the rumen microbiota in the buccal swab of pregnant Japanese Brown heifers. This experiment is interesting because it uses a challenging method. If published, this paper will add new information to the field of Japanese Brown cattle research.

General comments

This paper contains interesting experimental results and new information, so it is suitable for publication in Animals. However, there are some parts of the paper that should be explained in more scientific detail.

Specific comments

L2, Title: It should be specified that the rumen microbiota in the buccal swab samples.

L35-38: In which Table or Figure is this result shown?

L91-94: Having data for days 0 allows for better scientific interpretation. If you are sampling, please show the data.

L191-194: I thought this information should be included in the Materials and Methods.

Figure3: If there is a significant difference, please indicate it in the figure.

L227-237: Please indicate whether there is a statistically significant difference.

L241-249: Please indicate whether there is a statistically significant difference.

Figure6: I recommend showing separate heatmaps for each 30, 60 and 90 days.

L350-360: I recommend comparing the network analysis results of this experiment with other reports, such as those for rumen fluid samples.

L366-368: I thought that this text was not a conclusion drawn from experimental results.

Reviewer 3 Report

Comments and Suggestions for Authors

This study investigated the effects on the rumen microbiome of Japanese brown cattle fed with limonite. After reading this article, I realized that some parts of the article could be improved. Here are some of my suggestions for this article that I hope the authors will consider.

1. Line 30-32: The selection of the number of experimental animals was not consistent between the two groups, and the sample source was relatively small, which could lead to a large error in the experiment. There is no indication if the JBRK's are fed consistently, except with or without added limonite.

2. Line 84-86: Although the microbiota obtained from buccal swabs and rumen samples were very similar in a Pearson correlation analysis (Line 78-81), I believe that the rumen microbiota is much more complex than the buccal microbiota due to the complexity of feed digestion and metabolism by microorganisms in the rumen. Therefore, rumen samples should be collected for this experiment to be more rigorous.

3. Line 202-207: How were the ratios of oral and ruminal bacteria obtained? Is it correct to experimentally study the effect of limonite on the rumen microflora of JBRK, given that the ratios of oral and rumen bacteria in the two groups were different?

4. Line 311 (Discussion): I think your discussion is confused and does not clearly elucidate the mechanisms by which limonite has a beneficial effect on Japanese Brown cattle or the mechanisms by which the rumen microbiota changes.

5. Line 316-324: It is not necessary to state or discuss the rationale for the experimental approach taken here. Sampling directly from the rumen to study the rumen microbiome is universal and can be obtained by intubation without slaughtering the animal. If you want to demonstrate that buccal swab samples are adequate for rumen microbiome testing, you can collect rumen samples directly for comparison with buccal swab samples, which will also enrich your experiment and paper.

6. Line 325-335: What is the purpose of this paragraph, it doesn't seem to have much to do with your topic. You mention the benefits of feeding limonite on fatty acids of JBRK meat and body condition, but don't describe whether this was due to changes in rumen microbes.

7. Line 361(Conclusion): I suggest that you rewrite the conclusion. Make a summary of your entire article, clarifying what valuable information can be gleaned from the results of your experiment.

8. Line 393(Reference): Please check your references carefully. Some of the references are not properly cited and the page numbers are missing.

Reviewer 4 Report

Comments and Suggestions for Authors

Comments and Suggestions for Authors

After reviewing the manuscript entitled “Japanese Brown Cattle Fed with Limonite Exhibit Rumen Microbiome Differences”, the following suggestions were made it. The manuscript is generally poorly written and has several deficiencies, the main one being the low number of replicates used. The use of only four (and three for control) replicates in studies of this type makes the results obtained very unreliable, therefore, I consider that the study design does not meet the scientific rigor of a journal of high scientific prestige such as Animals. Furthermore, except for the materials and methods section, all sections of the manuscript are poorly focused and lack scientific soundness. The discussion and conclusions sections are probably the most problematic. Furthermore, the manuscript has important deficiencies in the introduction and data analysis. Therefore, I suggest rejecting the manuscript. Below are my specific comments:

Simple Summary

Lines 18, 21, and 26: Abbreviations should not be used in the Simple Summary.

Lines 23-26: This information is not clear.  The main findings are not clear. A brief and clear conclusion should be added at the end of the paragraph.

Abstract

Line 27: Authors should clearly write the objective of the study at the beginning of the Abstract.

Line 23: The number of replicates used (n=4 and n=3) is too low to test significant differences in continuous animal nutrition studies.

Lines 35-42: The information shown in these lines is not adequate. Additionally, all significant findings should be aggregated, and an appropriate significance value should be included whenever an effect or absence of effects is indicated.

Line 42: A clear and concise conclusion should be added at the end of the abstract. This is particularly important to help the reader understand the practical importance of the findings obtained.

Keywords: Japanese Brown cattle (JBRK); Aso limonite; rumen; microbiota. These words used as keywords are the same as those previously used in the title of the manuscript. Keywords should be different from those in the title (but related to the topic) to broaden the reach of academic search engines in case the manuscript is later published.

Introduction

Lines 47-72: The introduction does not clearly state the problem it seeks to solve. First, the authors must highlight what problem they are seeking to solve. Afterward, they must justify why they chose to study Limonite instead of another available product with similar effects. Likewise, these lines show that this is a study of regional importance, which makes it uninteresting for most readers. Finally, it is curious that the authors have only used four scientific references in two paragraphs. This shows that the literature review carried out on the problem studied was scarce. The introduction does not contain sufficient relevant references to support the importance of the study.

Material and methods

Lines 92 and 93: As mentioned above, four replicates per treatment (and three for control) are too low a number to be able to detect significant changes in studies of this type. Therefore, the results obtained are unreliable and lack scientific solidity.

Lines 172-176: The authors must indicate which statistical tests they used to evaluate the data's normality and homogeneity of variance. This is very important to know if the correct tests were used. Were repeated measures over time used in the analyses? This must be specified, and if repeated measures over time have not been used, the data must be reanalyzed using them to increase the statistical power of the analysis. Finally, the authors must add the specific statistical model used and justify based on the criteria they chose for the final models used. This evaluation is particularly important when analyzing data with repeated measures over time, as in the current study (days 30, 60, and 90).

Results

Lines 190-310: The description of the results is not clear and should be completely rewritten. Which treatment increased, decreased, or was not affected should be clearly stated and this description should be supported by the corresponding p-value. Likewise, the description should be done by groups of variables that showed similar behavior to avoid repeating the same information too many times. These same recommendations should be applied in the following descriptions of results shown in the manuscript. It is not enough to report treatment means and their standard errors or standard deviations; the observed p-value must be added.

Furthermore, because the data have a structure of equally spaced repeated measures over time, they should be analyzed that way, and the results should be reported as a single mean of treatment versus control. As a complement, the authors can also report the results for each measurement period, but only if significant period effects are detected.

Discussion

Lines 312-316: In these lines, the information previously shown in the introduction and material & methods sections is repeated. This is not suitable and should be removed.

Lines 317-323: In these lines, the importance of the buccal swab sampling method to evaluate rumen microbiota is discussed. Discussing the method is not correct since the study did not aim to compare methods. Therefore, these lines must be removed.

Lines 325-360:  La discusión realizada no está bien enfocada ni contiene la suficiente profundidad. Esta sección debería incluir explicaciones e hipótesis que clarifiquen cómo la Limonita modificó la microbiota. Estas explicaciones deberían incluir los mecanismos fisiológicos y bioquímicos de la Limonita. Sin embargo, los autores no realizaron esto. Por lo tanto, la sección de discusión debe reescribirse por completo.

Conclusion

Lines 362-371: The conclusions shown are not supported by the results found. In fact, this entire section needs to be revised. The authors use these lines to describe some of the findings obtained partially. Likewise, the authors provide some background on the livestock breeding system used in the current study. A good conclusion section should clearly show the study's contribution to the existing literature. Likewise, the conclusions must be in relation to the stated objective, the title of the manuscript, and the results obtained.

Comments on the Quality of English Language

The quality of English is acceptable
